# Conducting an Evaluation Framework of Importance-Performance Analysis for Sustainable Forest Management in a Rural Area

Hsing-Chih Chen [1], Tien-Pai Tseng [1], Kun Cheng [2], Supasit Sriarkarin [3], Wanyun Xu [2], Arockia E. J. Ferdin [1], Van Viet Nguyen [1], Cheng Zong [2] and Chun-Hung Lee [1,*]

[1] Department of Natural Resources and Environmental Studies, College of Environmental Studies, National Dong Hwa University, Hualien 97401, Taiwan; 810954001@gms.ndhu.edu.tw (H.-C.C.); 410054032@gms.ndhu.edu.tw (T.-P.T.); 810854010@gms.ndhu.edu.tw (A.E.J.F.); nvviet@vnuf2.edu.vn (V.V.N.)

[2] School of Wildlife and Protected Areas, Northeast Forestry University, Harbin 150040, China; chengkcn@163.com (K.C.); xwy414@163.com (W.X.); moredonkey@163.com (C.Z.)

[3] Department of Forest Management, Faculty of Forestry, Kasetsart University, Bangkok 10900, Thailand; supasit.sr@ku.th

\* Correspondence: chlee@gms.ndhu.edu.tw; Tel.: +886-3-8633343

**Abstract:** We established an evaluation framework for sustainable forest management (SFM) development based on locals' perspectives using the importance-performance analysis (IPA) method in a rural area of Taiwan. It identified the factors that affected local people's participation in and awareness of SFM based on local demographics, development factors of SFM, and perceptions of SFM, through the logistic regression method (LRM). Both the levels of importance and performance (I-P) of the SFM indicators were rated by the local residents and the differences between importance–performance among indigenous and non-indigenous people were examined. The factors that affected differentiation of local people's participation in the SFM program were: (1) forestry workers, (2) indigenous people, and (3) SFM development factors. The findings provide both theoretical constructs and policy implications for SFM mechanisms for the forest stewardship council (FSC) and sustainable development goals (SDGs) in a sustainable rural development.

**Keywords:** ecosystem services; social network; community participation; local people's awareness; sustainable development goals

## 1. Introduction

The importance of forest conservation and SFM in developing countries for sustainability goals has been highlighted [1], while management frameworks have also emphasized the importance of reducing deforestation rates, and conserving existing stocks of forest carbon using SFM [1,2]. In the context of the SDGs, there is increasing emphasis on technologies and approaches that have raised ambitions for the sustainable governance of forests. With respect to effectiveness and equitability, conventional approaches are, at the same time, focusing on forest resource governance related to establishing and protecting private property rights, creating markets, and mobilizing private finance, which are underlying drivers of deforestation [3–5]. The conflict has arisen due to the multiple understandings of biodiversity conservation and sustainable management by the various stakeholders [6,7]. Thus, from a long-term forest management interventions process, SFM is focused on forest resource evaluation relating to the productive, protective, and social roles of forest ecosystems, monitored regularly over time [8].

Community participation is regarded as one of the key factors for effective forest governance in tropical countries [9]; there is a consensus that environmental problems should not be addressed by governments in isolation but also need public participation in environmental governance [10,11]. However, any negotiation or theory is more difficult

than its practical application [3]. In recent years, the traditional land, hunting, and gathering rights of indigenous people have been controversial [12–15]. It has been observed that some local people do not support the sustainable development of forest resources at their disposal. However, community forest management (CFM) is a form of forest management that encourages local people to participate in management with forest management officials under the framework of SFM [16]. Relevant research on local tenants and indigenous traditional cultural knowledge may also help policy planning and improve the relationship between stakeholders [17]. Identifying the factors influencing public participation in decision-making on SFM programs, and the implementation of actions related to SFM, is critical to ensuring a greater impact on SDGs and well-being.

Rural development and community participation are essential factors for improving SFM. The participatory approach focuses on the collaboration of individuals and local government institutes to promote the awareness of local people and to create potential opportunities for the execution of local government programs [18]. Integrating rural development, community participation, and local awareness of the forest environment, can provide essential perspectives, and assist in devising an effective program for SFM development. These principles may also be seen as identifying, and responding, to the viewpoints of indigenous and non-indigenous people concerning resources, which is a critical indicator for the success and sustainable development of SFM [12,19]. The different perspectives of indigenous and non-indigenous people concerning the outcomes of SFM can provide criteria that may be integrated into the SFM evaluation framework. From this point of view, it is important for local managers to reflect the main perspectives in proposals for SFM development in a rural area.

With respect to the varying bases for individual satisfaction with and evaluation of goods or services, the IPA method has been successfully utilized to apply evaluation frameworks to capture consumer satisfaction, based on a series of indicators provided from target strategies [20,21]. The IPA method identifies and integrates participants' perceptions of I-P indicators of a target into a two-dimensional grid arranged in four quadrants, so that the program leader can devote resources and design management strategies relevant to the "concentrate and main quadrant" based on the stakeholders' understanding of what has been explicitly plotted [20,22–24]. In the past two decades, the IPA method has been employed in education [25], health care [26], ecosystem services [20,27,28], and hospitality and tourism [24,29–31].

To the best of our knowledge, no studies to date have simultaneously evaluated locals' opinions of the I-P dimensions with respect to their perception of policy effects using both primary indicators of SFM and the different principles of the FSC [12,14,32], even though the IPA method has been used successfully in the fields of environmental management [33,34], and natural resource management (NRM) [20,35]. Evidently, a lack of systematic evaluation exists for the conduct of socially, environmentally, and economically relevant SFM strategies in communities from rural areas. This paper first establishes an evaluation framework for the development of SFM, based on indigenous and non-indigenous resident perspectives, using the IPA method, in a rural area of Taiwan. The framework aims to identify the factors that affect locals' participation awareness for the SFM, based on local people's demographics, the development principles of SFM, and perceptions of SFM through the LRM. This study, in Section 2, provides a systematic synthesis of community forest management, rural development, sustainable forest management, and FSC principles, with respect to participation behavior and SFM. Then, we establish the conceptual framework in Section 3, including the study area, research method, and research design, based on the above study purposes. Section 4 provides the empirical outcomes of the matrix of the importance-performance levels of the SFM indicators in a rural area, the SFM relevance of the eight indicators and participation behavior, and local peoples' participation behavior models of SFM. Finally, in Section 5, we synthesise theoretical constructs and policy implications for the SFM process and for the FSC in sustainable rural development.

## 2. Literature Review

### 2.1. Community Forest Management and Rural Development

CFM has been shown to be an effective tool for the improvement of forest management, with respect to environmental and socio-economic outcomes, compared to existing alternative management regimes [36–39]. CFM has been applied, as the pathway and alternative strategy, to stimulate the planning and fulfillment of the 'reducing emissions from deforestation and forest degradation plus' (REDD+) scheme [39]. CFM is also a method of forest management which permits local residents to work together with the forest manager and to focus on the issue of sustainable NRM [16,40], and is a method which addresses quality of life for local communities under the conceptual framework of SFM [40]. CFM explicitly refers to forest resource use, and regional governance defining local peoples' rights, liabilities, and authority for SFM within local communities [9,40].

Mihai & Iatu [41] pointed out that rural area development importantly integrates the factors of social, demographic, economic, governance, and environmental dimensions, with land use management also a main perspective for future rural development perspectives under the CFM framework [41]. Local people's participation in CFM needs to be conducted in an atmosphere of mutual trust, emphasizing the residents' perceptions of SFM, and stimulating suitable SFM realities by focusing on locals' involvement in policy-making [40]. In addition, CFM also relates to the collective participation of all potential stakeholders being involved with forest management issues and producing SFM outcomes [10], and emphasizes the requirement for local participation in environmental governance [10,11,40]. As a result, participation in NRM programs by the community brings benefits to them in solving rural development problems and improving community wellbeing [42]. The elements of institutional cooperation, fair benefit-sharing mechanisms, and higher community ability for monitoring, reporting, and confirmation, are the important factors for successful CFM development [39], enabling CFM to contribute to the SDGs for sustainable agriculture [41]. To sum up, it is important to integrate CFM into rural area development.

### 2.2. Sustainable Forest Management and the FSC

Public participation enhances locals' involvement in SFM. It highlights the need for content-specific methods to ensure community engagement in forest management programs for policymaking [40]. It is essential for forestry companies and local communities to ensure community participation and consultation with multiple stakeholders under the criteria of the FSC [12,32]. According to the principles of the FSC, forest certification applies a market-based process, using an understanding of forest resources, in order to achieve the goal of SFM [12,43]; the certification of forest management can help reinforce local people's willingness to participate in SFM [12]. Therefore, it is critical to apply indicators of SFM, under the auspices of the FSC, in rural area development, which include workers' rights, indigenous peoples' rights, community relations, environmental values, and conservation values [12,39,40,44–47].

With respect to social development in a rural area, Boron [44] suggested that social development should focus on improving security and human rights for the local communities, especially with respect to worker's benefits and rights [44]. As a consequence, there will be a positive effect on security and human rights from the perspectives of institutional presence, monitoring, and enforcement [44]. Degnet [12] demonstrated that property rights are a key factor for plantation activities with villagers in developing countries, with land rights and employment instability the main factors affecting community engagement under plantation forest management [12,14]. Local communities also have a significant influence from the perspective of the economy and environmental values under a forest plantation program. This relates their economies and rights to management activities occurring adjacent to a rural area [32]. FSC certification also addresses indigenous people's rights for community participation in SFM [14,15], impact assessment, and monitoring process [45], requiring forest owners to respect indigenous people's rights [12,13].

Community participation is a critical factor for SFM, affecting the outcomes of forest resources management, with community relations, understanding of community perceptions, and participation in programs and activities central [12]. It can also support NRM and bring benefits to communities in reducing environmental problems and improving community wellbeing [42]. A high level of interaction between a local manager and local communities, regarding both tangible and intangible resources, can have the positive effects of more profound and vital community relations [12,39,48], as demonstrated by organizations and groups in Argentina, Brazil, Chile, Nepal, Thailand, and Vietnam [12,39,45,47,49]. In contrast to rural economic and social development, conservation of biodiversity and natural resources is the foundation of SFM [44]. Furthermore, a robust regulatory and monitoring framework has been shown to be a successful method for biodiversity conservation in productive landscapes and rural areas [44,50–52]. To sum up, preserving the resources of higher cultural value, respecting local needs, and monitoring the impacts of forest management, are essential. Impacts on the local environment are the main issues for the FSC concerning the principles of "environmental values and impact" and "high conservation values", which are essential concepts for the integration of SFM into rural area development. The need to apply an evaluation framework for sustainable forest management, and to access local people's perceptions towards policy effects, under SFM and FSC certification, has become a key issue.

### 2.3. Participation Behavior and SFM

In terms of pathways towards SFM, many studies have pointed out that community consensus is key. Local communities' participation is a key variable for sustainable management within the framework of SFM [19,45]. Public participation in environmental protection requires seeking involvement of the community who rely on the natural resources [45,53]. Further, communities with a more positive attitude to long-term governance are more likely to adopt the protective actions, since attitude is a primary factor for sustainable behavior in SFM [3,54–56]. Obviously, different communities vary in participatory barriers dependent on their socioeconomic structure [57], which are key issue when exploring the factors influencing community participation in SFM. Financial support is also key to SFM [3]. Savari [19] pointed out that wellbeing depends on timbered and non-timbered incomes, and the indirect use of forest resources from forest management. A manager may be able to find ways to reduce community dependence on forest resources by raising communities' income in other areas [58]. Thus, reducing native dependency on forest resources is a prerequisite to adopting SFM. Many previous studies have also proposed decreasing a community's dependency on forest resources by providing non-forest employment [3,19].

Evaluation of psychological dimensions has revealed that enhancing a community's knowledge of NRM is a key factor with a positive influence on SFM [3,59,60]. This requires the raising of community consensus [19,45,53,57], financial support [19,58], human capital [19,59,60], and capacity building [19,59]. The research suggests that it is important to have regular training and workshops to build the capacity of local communities, to train community residents as nature contributors, to develop them for the work on the monitoring of natural resources, and to let them recognize the consequences and impacts of forest degradation for the SFM [3,19,41,45,59]. The path towards sustainable development requires enhancing public education on natural resources protection since sustainable and inclusive development requires sustained motivation [46,61]. An integrated education system, established for the local community about SFM, would benefit the optimized utilization of natural resources [3,59–61]. SFM cannot be achieved without adequate education for the rural population [3,8,41]. Therefore, identifying the factors that affect local people's participation and awareness of SFM, based on local demographics and development factors, are key issues for policy-making under an SFM framework.

## 3. Conceptual Framework

### 3.1. Study Area

Taiwan's government began to implement the "extension of the afforestation policy" in 2007 [62], with the goals of SFM, and community-based tourism [63,64]. This captured local people's perspectives, to address the growing level of environmental awareness, under the framework of SFM, and the principles of FSC certification, in forest parks and rural areas [62,63,65–67]. To promote environmental and natural conservation with carbon reduction, Taiwan's government announced that both the public and private sectors can enhance ecological communities by means of SFM [65,67–70]. Thus, the above issues can be a basis for the evaluation of the achievements of SFM [62,71,72].

From the perspectives of domestic environmental groups and international initiatives, the Forestry Bureau, and relevant government regulations, recognize that forestry policy would have to adapt to the SFM and FSC programs [65,71]. The Forestry Bureau has attempted to solve any negative impacts of third-party forest certification based on FSC principles [65]. The forestry companies understand that the government sector will recognize certified forestlands as a symbol of good forestry practice using FSC guidelines [65,73]. Forest certification is an ongoing process in Taiwan, with a primary focus on logging and bamboo cutting [65], while the evaluation of workers' and indigenous peoples' rights [74–77], community relations [72,78], environmental values, and conservation values [62,78–80], are all lacking.

Hualien County, which is located in the east coast of Taiwan (Figure 1), was selected as the study area because of its many cultures among indigenous groups, its natural resources, and its beautiful landscapes [62–64,67]. This county is rich in forest resources, and government units almost completely control the ownership of its forest land [81]. From north to south, the forest land in the district is divided into five business areas: Liwu River, Papaya Mountain, Lintian Mountain, Yuli, and Xiuguluan. The forest area of Hualien County is 318,580 hectares, accounting for about 20.77% of the total forest area in Taiwan [82]. After discussion with the staff of the Forest Management Office, and data compilation, the survey selected four villages for analysis around the forest thinning operation area, between 2018 and 2020. In the Lintian Mountain business area, the total forest area is 66,438 hectares, and there are 151 forest lands in total. This research is located in the 125–127 forest group in this business area, between 8 and 24 kilometers of the Guangfu forest road. The area is about 175 hectares in total, and the altitude ranges from 200 m to 1500 m. The affected administrative districts are the Mingli Village and Dama Village in the Guangfu township.

According to census data in December 2019, the population of Mingli Village was 801 people (278 households), of which 750 were indigenous people; the population of Dama Village was 1132 people (468 households), of which 794 were indigenous people. In the Yuli business area, the total forest area is 57,704 hectares, divided into 103 forest lands. This study is located in the 25–28 forest group, 10.2 km to 19.5 km from the Ruisui forest road, and from 0 km to 3.8 km from the Sanmin forest road. The area is about 132 hectares, and the altitude ranges from 1100 m to 1500 m. The affected administrative regions are Hongye Village in Wanrung Township and Ruixiang Village in Ruisui Township. There were 1257 people (437 households) in Hongye Village, of which 1198 were indigenous people; the population of Ruixiang Village was 1078 people (499 households), of which 281 were indigenous people [83].

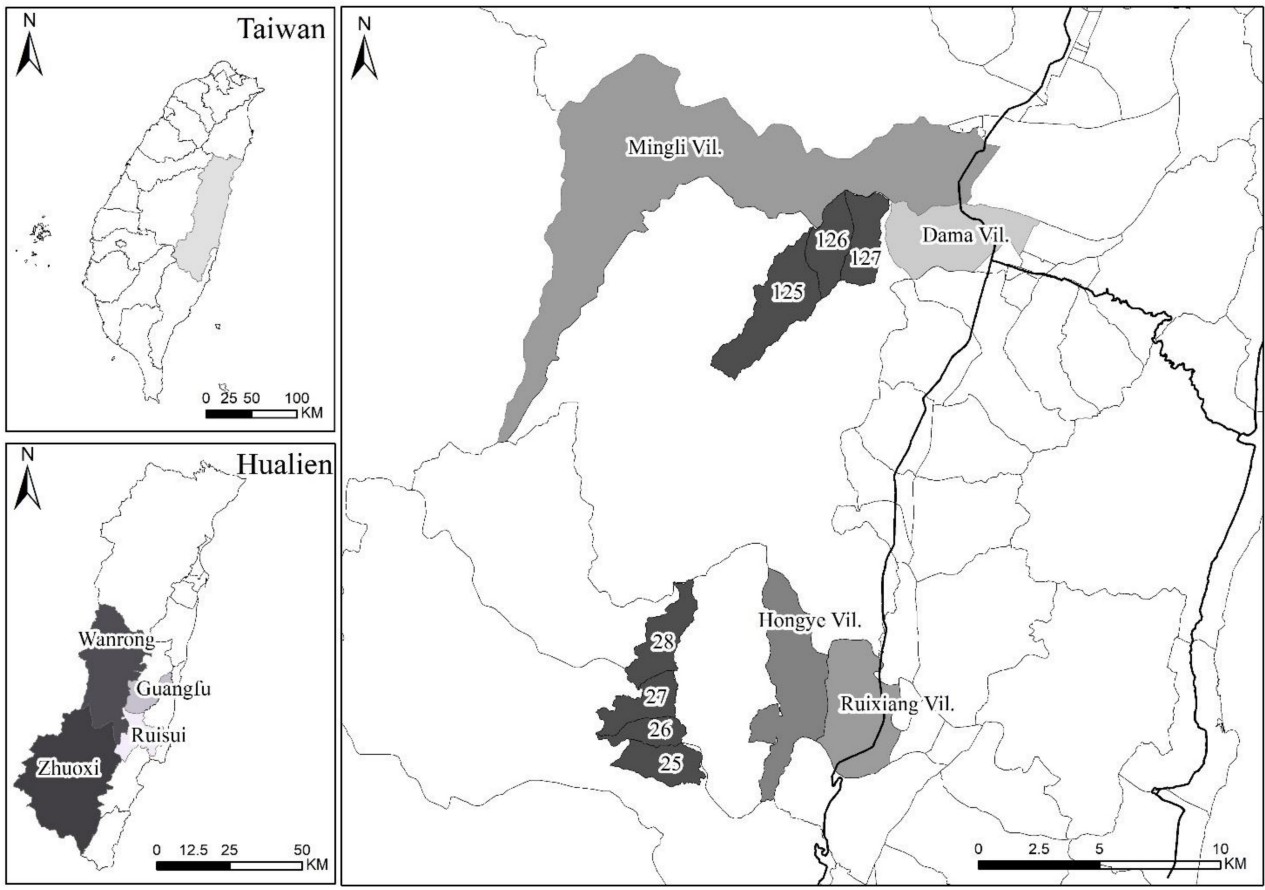

**Figure 1.** The study area and sampling sites.

### 3.2. Research Method

Past IPA studies have primarily asked respondents to talk about their awareness of agreement regarding particular topics [84]. The IPA has not provided an extra level of contextualization to respondents' attitude study by asking them to estimate the extent to which they agree or disagree with a statement about an impact, but rather by analyzing respondents' perceptions of both importance and performance statements [20,29]. Martilla and James [85] proposed an IPA, which was based on the customer's point of view and evaluated the products or services provided by a company. This research approach has been widely used in various fields over time [20,22–29]. By establishing indicators for products and services before service (importance), and after service (performance), relevant causal relationships can be established [23]. Through analysis of causality, it is possible, using the IPA method, to consider the educational contribution of intangible products [86], or to use it with other research methods to explore sustainable tourism development from residents' perspectives [31].

This study applied the IPA to estimate the importance of SFM to communities, and perceptions of the performance of the current forest management system. We combined FSC indicators and, after two interviews with colleagues in the Hualien Forest Management Office, a total of eight "social aspects" and "environmental aspects" were defined. Firstly, group indicators were utilized to measure the perception of SFM. The interviewees were asked to rate the attributes in two respects, their "importance" and their "performance." "Importance" refers to the respondents' preference and emphasis regarding SFM, while "performance" refers to the actual performance of the forest management system. The original values of the eight groups of indicators collected by the survey were averaged, according to the classification, and then converted into coordinate values with standardized

values (Z-score), with importance as the horizontal axis, and expressiveness as the vertical axis. In addition, the coordinate values were plotted on the plane coordinates.

The four quadrants of the IPA have the following meaning [24]: Quadrant A indicates that both importance and performance are high, and the attributes falling in this quadrant should continue to be maintained ("keep up the good work"), which can also be represented as "strength"; Quadrant B indicates high importance but poor performance. Attributes falling in this quadrant should focus on improvement ("concentrate here") and can also be represented as "threats"; Quadrant C indicates that both importance and performance levels are low. The falling attributes of this quadrant usually imply that they are of lower priority for improvement, and can be represented as "weakness"; Finally, quadrant D means that the importance is low, but performance is high. The rated attributes imply possible "overkill", though, if there is a good reason to continue investing resources, it can also be represented as "opportunities" (Figure 2).

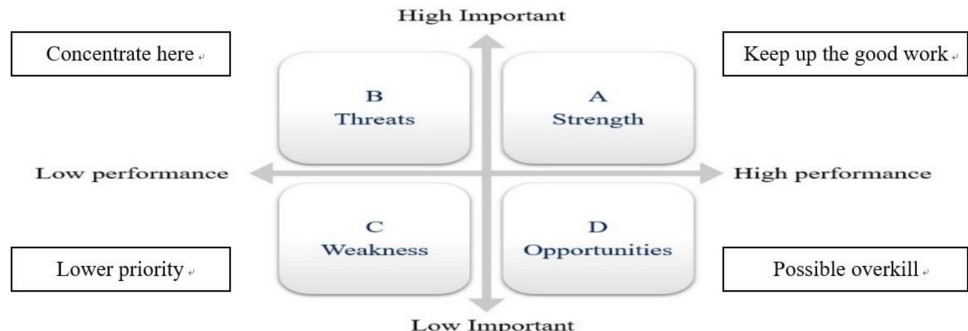

**Figure 2.** Evaluation framework for importance and performance analysis (Revised from [24,27]).

Applying binary choice theory, the logistic regression method (LRM) was used to examine locals' participation behavior with respect to SFM [87]. In the LRM, the variable of local people's participation awareness for SFM was the dependent variable; 1 represents prefer to join the SFM, 0 for another. The independent variables included demographic information (i.e., gender, marital status, forestry worker, and ethnic group), and psychological and attitudinal dimensions regarding SFM (i.e., community consensus, financial support, human capital, and capacity building). The demographic, psychological, and attitudinal dimensions are dummy variables, and mean importance and mean performance measures are quantitative variables. Two models were built in this study. The first model, had local people's participation awareness for SFM as the dependent variable while other factors on SFM were excluded, with the overall importance mean of SFM indicators as the independent variables. We performed the comparisons between logit and probit regression analysis, applying binary choice theory [87,88]. Finally, the model's goodness of fit (GOF) was evaluated by the Akaike information criterion (AIC) and log-likelihood ratio (LLR) [78–80,87,88].

### 3.3. Research Design

An SFM index corresponding to the FSC principles was firstly developed based on empirical studies, which firstly included the workers' rights and employment conditions [44,89], indigenous peoples' rights [12–15,32], community relations [12,32,39,40,42,47,48], environmental values and impact [12,14,44,45], and high conservation values [44,50–52]. Secondly, focus group discussions were used to capture the stakeholders' perspectives (i.e., forest owners, direct interests, and indirect interests) [90]. Finally, the formal questionnaire design was developed based on the discussion groups and the pre-test questionnaire survey. FSC is an independent worldwide non-government organization (NGO) that conducts the standards and predefined indicators for SFM, to boost the social, economic, and environmental outcomes of forest resources [14]. The SFM indicators were extracted and adapted from the literature review and FSC principles, respectively, for local communities.

In this study, in combination with the FSC principles, a total of eight SFM indicators were classified and presented (Table 1).

**Table 1.** Sustainable forest management (SFM) index corresponding to the Forest Stewardship Council (FSC) principles.

| No. | Indicators of the SFM (Abbreviation) | Principles of the FSC | Literature |
|---|---|---|---|
| A | Enhancing the workers' rights for their forest management work (WORKERRIGHT) | Workers' rights and employment conditions (Principle 2) | [44,89] |
| B | Holding the SFM and thinning meetings and negotiating the local's concern issues proactively (NEGOTIATION) | Indigenous peoples' rights (Principle 3) | [12,14,15] |
| C | Providing the locals' rights from the hunting and gathering of the forest resources (HUNTINGRIGHT) | Indigenous peoples' rights (Principle 3) | [12,13,32] |
| D | Establishing the community patrol organizations for the forest management (ORGANIZATION) | Community relations (Principle 4) | [32,40,47] |
| E | Providing the social and economic benefits for the local community (BENEFITS) | Community relations (Principle 4) | [12,32,39,42,48] |
| F | Providing the water resources for the local needs (WATERRESOURCES) | Environmental values and impact (Principle 6) | [12,14] |
| G | Monitoring the impacts of the SFM and thinning operations on local environment (MONITOR) | Environmental values and impact (Principle 6) | [12,44,45] |
| H | Preserving the resources of higher cultural values and local needs (PRESERVE) | High conservation values(Principle 9) | [44,50–52] |

* Adapted from FSC (2015), and stakeholder interviews of this study.

A survey method was utilized for the data collection [24], by means of questionnaires for local residents in a rural area of Taiwan, plus face-to-face interviews. This study incorporated three questionnaire elements. The first addressed local people's awareness of development factors in SFM. The second part comprised a list of five-point Likert scales for evaluating the importance and performance of the SFM development indicators in Taiwan. These were rated from "very important/strongly satisfied" to "very unimportant/strongly dissatisfied" [24] (Table 1). Finally, we included the respondents' demographic details (i.e., gender, marital status, forest worker, and ethnic group), as well as socio-economic information. The respondents were asked to evaluate their importance awareness regarding 8 SFMs on five-point Likert scales (from "1—very unimportant" to "5—very important"). A description of our topic was then provided verbally by the on-site interviewers, and the respondents' interactions with SFM (such as their opinions on forest management, major concerns on FSC awareness, awareness, etc.) were probed. This was designed to help enable respondents to establish an emotional and behavioral link with the current status of forests [20,27,28,91]. The interviews were managed by well-trained scholars at the four sites selected between June 2019 and April 2020 (Figure 1), using a stratified random-sampling procedure.

## 4. Results and Analysis

The formal on-site survey for the local residents was undertaken between July 2019 and March 2020 at our research site including the Mingli, Dama, Hongye, and Ruixiang villages, surrounding the Gungfu and Ruisui township (Figure 1). Based on a confidence level of 95%, and estimation bias of 5%, we assumed that the indigenous and non-indigenous

people had the same preference for the SFM program. We collected a total of 384 samples. In total, we obtained a total of 400 resident responses; a summary of the frequencies and percentages for their demographics is shown on Table 2. The research surveys were in four villages that included Hongye Village, Ruixiang Village, Mingli Village, and Dama Village. The actual samples for the villages were 115 (28.75%), 106 (26.5%), 71 (17.75%), and 108 (27.0%), respectively. The gender ratio was nearly equally divided (half male and half female), with the largest percentage married (86.25%). The largest group in terms of age were elders, 60 years and above (35.75%), followed by those aged 50–59 years (24.25%), and then those aged 40–49 years (18.50%). Most of the villagers had education lower than university level, with junior high school or lower (49.25%), and junior high school (35.75%). Almost half of the respondents had a monthly income lower than 20,000 new Taiwan dollars (NTD) (Table 2).

**Table 2.** Definition of Variables in the Residents' Socio-economic Background Information.

| Variables | Level | All Residents (*n* = 400) | | Non-Indigenous Residents (*n* = 134) | | Indigenous Residents (*n* = 266) | |
|---|---|---|---|---|---|---|---|
| | | Frequency | % | Frequency | % | Frequency | % |
| Site | Hongye Village | 115 | 28.75 | 10 | 7.46 | 105 | 39.48 |
| | Ruixiang Village | 106 | 26.50 | 81 | 60.45 | 25 | 9.40 |
| | Mingli Village | 71 | 17.75 | 3 | 2.24 | 68 | 25.56 |
| | Dama village | 108 | 27.00 | 40 | 29.85 | 68 | 25.56 |
| Gender | Female | 206 | 51.50 | 59 | 44.03 | 147 | 55.26 |
| | Male | 194 | 48.50 | 75 | 55.97 | 119 | 44.74 |
| Marry | Single | 55 | 13.75 | 14 | 10.45 | 41 | 15.41 |
| | Married | 345 | 86.25 | 120 | 89.55 | 225 | 84.59 |
| Age (years) | 20–29 | 47 | 11.75 | 4 | 2.99 | 43 | 16.17 |
| | 30–39 | 39 | 9.75 | 10 | 7.46 | 29 | 10.90 |
| | 40–49 | 74 | 18.50 | 23 | 17.16 | 51 | 19.17 |
| | 50–59 | 97 | 24.25 | 38 | 28.36 | 59 | 22.18 |
| | 60 or above | 143 | 35.75 | 59 | 44.03 | 84 | 31.58 |
| Education | Junior high or below | 197 | 49.25 | 61 | 45.52 | 136 | 51.13 |
| | Senior high | 143 | 35.75 | 47 | 35.08 | 96 | 36.09 |
| | University or above | 60 | 15.00 | 26 | 19.40 | 34 | 12.78 |
| Income (NT$/month) | 20,000 or below | 194 | 48.50 | 65 | 48.51 | 129 | 48.50 |
| | 20,001–30,000 | 71 | 17.75 | 26 | 19.40 | 45 | 16.92 |
| | 30,001–40,000 | 51 | 12.75 | 11 | 8.21 | 40 | 15.03 |
| | 40,001–50,000 | 47 | 11.75 | 17 | 12.69 | 30 | 11.28 |
| | 50,001 or above | 37 | 9.25 | 15 | 11.19 | 22 | 8.27 |

### 4.1. Matrix of the I–P Levels of SFM Indicators in a Rural Area

This study examined the SFM perspectives from indigenous and non-indigenous residents using a paired-sample *t*-test, using an evaluation framework for SFM indicator development in an IPA (Table 1) in Table 3. We summarize the mean scores for all indicators and present the top and the bottom for three ranks of SFM indicators for indigenous and non-indigenous residents, respectively. The importance level of most SFM indicators is significantly larger than the performance level at a 10% significant level, with a difference of over 0.23 points (Table 3).

Non-indigenous people consider that the SFM indicators about environmental values and impact [item 7], workers' rights and employment conditions [item 1], environmental values and impact [item 6], and indigenous peoples' rights [item 2], are more important, across all the SFM indicators. Regarding SFM performance, most individual indicator perspectives were found to outperform the other indicators except for the indicator, "providing the locals' rights from the hunting and gathering of the forest resources". The indigenous peoples' rights, community relations, and high conservation values of SFM development are all relatively unimportant, and performed poorly [items 3, 5, 8]. Moreover, residents

consider it to be unimportant to provide the local people with rights from the hunting and gathering of the forest resources (mean = 2.75), while providing social and economic benefits for the local community (mean = 3. 2), and preserving the resources of higher cultural value and local needs in the SFM decision-making process (mean = 3.89).

**Table 3.** Mean scores and paired-sample *t*-test among residents for the SFM indicators on importance and performance levels.

| No. | Indicator | Importance Mean (Rank) | Performance Mean (Rank) | Difference | t-Value | *p*-Value |
|---|---|---|---|---|---|---|
| | | All residents (*n* = 400) | | | | |
| A | WORKERRIGHT | 4.20(3) | 3.44(2) | 0.76 | 11.86 | 0.0001 |
| B | NEGOTIATION | 4.36(2) | 2.89(8) | 1.47 | 17.25 | 0.0001 |
| C | HUNTINGRIGHT | 3.60(7) | 2.97(7) | 0.63 | 7.25 | 0.0001 |
| D | ORGANIZATION | 4.01 | 3.39(3) | 0.62 | 10.08 | 0.0001 |
| E | BENEFITS | 3.39(8) | 2.98(6) | 0.41 | 6.09 | 0.0001 |
| F | WATERRESOURCES | 4.17 | 3.91(1) | 0.26 | 5.38 | 0.0001 |
| G | MONITOR | 4.44(1) | 3.29 | 1.15 | 15.97 | 0.0001 |
| H | PRESERVE | 3.94(6) | 3.18 | 0.76 | 11.52 | 0.0001 |
| | Overall mean | 4.01 | 3.26 | | | |
| | | Non-indigenous residents (*n* = 134) | | | | |
| A | WORKERRIGHT | 4.22(2) | 3.50(2) | 0.72 | 7.23 | 0.0001 |
| B | NEGOTIATION | 4.01 | 2.74(8) | 1.27 | 8.43 | 0.0001 |
| C | HUNTINGRIGHT | 2.75(8) | 3.16(6) | -0.41 | −3.28 | 0.0010 |
| D | ORGANIZATION | 3.98 | 3.44(3) | 0.54 | 5.13 | 0.0001 |
| E | BENEFITS | 3.20(7) | 2.97(7) | 0.23 | 1.84 | 0.0680 |
| F | WATERRESOURCES | 4.13(3) | 3.97(1) | 0.16 | 2.00 | 0.0480 |
| G | MONITOR | 4.49(1) | 3.37 | 1.12 | 8.79 | 0.0001 |
| H | PRESERVE | 3.89(6) | 3.19 | 0.70 | 6.15 | 0.0001 |
| | Overall mean | 3.83 | 3.29 | | | |
| | | Indigenous residnets (*n* = 266) | | | | |
| A | WORKERRIGHT | 4.18 | 3.41(2) | 0.77 | 9.50 | 0.0001 |
| B | NEGOTIATION | 4.53(1) | 2.98(6) | 1.55 | 15.28 | 0.0001 |
| C | HUNTINGRIGHT | 4.03(6) | 2.88(8) | 1.15 | 11.85 | 0.0001 |
| D | ORGANIZATION | 4.03(6) | 3.36(3) | 0.67 | 8.87 | 0.0001 |
| E | BENEFITS | 3.48(8) | 2.97(7) | 0.51 | 6.38 | 0.0001 |
| F | WATERRESOURCES | 4.19(3) | 3.88(1) | 0.31 | 5.15 | 0.0001 |
| G | MONITOR | 4.40(2) | 3.25 | 1.15 | 13.44 | 0.0001 |
| H | PRESERVE | 3.97(7) | 3.18 | 0.79 | 9.78 | 0.0001 |
| | Overall mean | 4.10 | 3.24 | | | |

Indigenous residents, having the same set of SFM indicators, are concerned more about the indicators relating to indigenous peoples' rights (mean = 4.53 for item 2), and the environmental values and impact (items 6, 7). The indicator of "enhancing the workers' rights for their forest management work" (mean = 4.18) also is outstanding in the importance list. Indigenous residents perceive it to be unimportant to provide social and economic benefits for the local community (mean = 3.48), as well as preserving the resources of higher cultural value and local needs in the SFM decision-making process (mean = 3.97). However, indigenous residents perceive indigenous peoples' rights [items 2, 3] and community relations [item 5] on SFM as relatively unimportant as shown by their poor performance. Comparing the SFM performance, indigenous residents recognize that all individual indicators of SFM importance outperform the other indicators.

### 4.2. SFM of the 8 Indicators and Participation Behavior

From the indicators of SFM, the eight indicators of I–P level were analyzed and summarized, and all the interviews of the indigenous and non-indigenous groups were discussed and compared (Figure 3). Based on the overall comparison results, the interviewed residents' perceptions of SFM and IPA, relating to enhancing the workers' rights

for their forest management work, providing the water resources for local needs, and monitoring the impacts of the SFM and thinning operations on local environment, three projects (such as "Team") were located in the A quadrant from the perspectives of indigenous and non-indigenous groups. Therefore, local governance should continue to maintain these advantages and meet the SFM goals with the FSCs' principles. In addition, the project relating to holding the SFM and thinning meetings, and negotiating the locals' concern issues proactively, was located in the B quadrant. Evidently, most respondents believe that there is room for improvement in the performance of this project, and that the management should be strengthened and improved in the future. The views of all respondents on providing the locals' rights from the hunting and gathering of the forest resources and providing the social and economic benefits for the local community, are in the C quadrant, which means that it is "current SFM." The above are low priority items for SFM in rural areas.

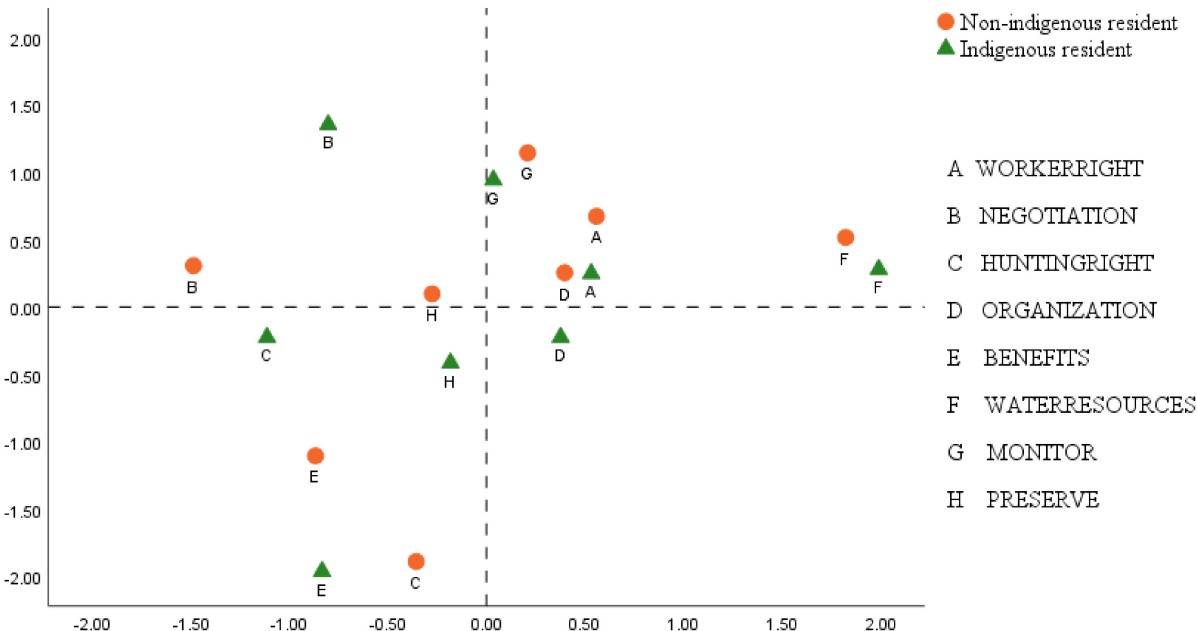

**Figure 3.** Differences in overall and non-indigenous and indigenous residents' perceptions of the 8 indicators of SFM indicators.

Based on the views of non-indigenous residents, establishing community patrol organizations for forest management is located in quadrant A. The non-indigenous residents would like to keep the advantages of the SFM goals under the "community relations" principle of the FSC, while preserving the resources of higher cultural value, and local needs; these are located in quadrant B. The non-indigenous residents recognize that management ought to focus on improving the "high conservation values" principle of the FSC. Relatively, preserving the resources of higher cultural value and local needs are located in quadrant C. The establishment of community patrol organizations for forest management is located in quadrant D. From the perspective of indigenous residents, these outcomes show that "high conservation values" have a lower priority in the SFM indicators and need to be improved, while "community relations" is the main factor for SFM development in the future (Figure 3).

*4.3. Local People's Participation Behavior Models of SFM*

We used local people's participation awareness for SFM in Taiwan's rural areas as the dependent variable (LPSFM), and considered demographic details (i.e., gender, marital status, forest worker, and ethnic group), development factors on SFM (i.e., community consensus, financial support, human capital, and capacity building), and the overall importance mean of SFM indicators, as the independent variables. Similarly, the second model

contained the overall performance mean of SFM indicators, while all the other demographic and development factors on SFM as the independent variables were integrated in the logit and probit regression models, based on the above dependent and independent variables, respectively [87] (Table 4).

**Table 4.** Estimation results of local people's perception for SFM participation behavior.

| Variable Names | Logit Model | | Probit Model | |
|---|---|---|---|---|
| | Importance on SFM (Model I) | Performance on SFM (Model II) | Importance on SFM (Model I) | Performance on SFM (Model II) |
| | Coeff. (Std error) | Coeff. (Std error) | Coeff. (Std error) | Coeff. (Std error) |
| Constant | −8.140 *** (1.293) | −2.725 *** (0.801) | −4.781 *** (0.728) | −1.660 *** (0.472) |
| Gender (1 represents male, otherwise is 0) | 0.459 * (0.252) | 0.413 * (0.243) | 0.263 * (0.148) | 0.236 (0.144) |
| Marital status (1 represents marriage, otherwise is 0) | 0.757 ** (0.372) | 0.671 * (0.365) | 0.457 ** (0.219) | 0.404 * (0.214) |
| Forestry worker (1 belongs forestry worker, otherwise is 0) | 0.588 ** (0.272) | 0.796 *** (0.261) | 0.343 ** (0.162) | 0.472 *** (0.157) |
| Ethnic group (0 represents indigenous people, otherwise is 1) | −0.452 (0.283) | −0.736 *** (0.272) | −0.279 * (0.165) | −0.431 *** (0.157) |
| Community consensus (1 means agree this factor on SFM, otherwise is 0) | 0.313 (0.252) | 0.401 * (0.243) | 0.196 (0.148) | 0.265 * (0.144) |
| Financial support (1 means agree this factor on SFM, otherwise is 0) | 0.597 ** (0.292) | 0.635 ** (0.293) | 0.355 ** (0.176) | 0.387 ** (0.177 |
| Human capital (1 means agree this factor on SFM, otherwise is 0) | 0.598 ** (0.262) | 0.733 *** (0.254) | 0.351 ** (0.155) | 0.444 *** (0.150) |
| Capacity building(1 means agree this factor on SFM, otherwise is 0) | 0.895 *** (0.306) | 0.973 *** (0.299) | 0.545 *** (0.182) | 0.594 *** (0.178) |
| Mean Importance | 1.409 *** (0.285) | - | 0.822 *** (0.163) | - |
| Mean Performance | - | 0.120 (0.204) | - | 0.073 (0.121) |
| AIC | 425.8 | 453.2 | 425.2 | 452.4 |
| AIC/N | 1.065 | 1.133 | 1.063 | 1.131 |
| LLR | 89.45 | 62.087 | 90.098 | 62.909 |
| Chi square value | $\chi^2(9, 0.01) = 21.67$ | | | |

***, **, * are significance at $p < 0.01$, $p < 0.05$, and $p < 0.1$, respectively.

In model I, most variables were positively correlated with local people's SFM participation behavior, which was consistent with the logit and probit regression models. This indicates that the respondents will join local SFM programs, which contain the various groupings of males, marital status, forestry workers, and those with a higher perception of the importance of SFM, and recognizes that SFM development needs the indicators of financial support, human capital, and capacity building. For model II, we found that respondents will join the local SFM program, which contains the groupings of males, marital status, forestry workers, and indigenous people, and which acknowledges that SFM development has to integrate the factors of community consensus, financial support, human capital, and capacity building. The results are also consistent with the two binary choice model (Table 4). The GOF of our empirical model also meets the criterion of the LLR and AIC [88], which shows that the local people's participation behavior models of

SFM had solid results with model specifications under the binary choice model and the normality of probit model, with adequate sample size [78–80,87,88].

Thus, for SFM participation, a local manager may design the platform, and encourage local residents, by gender, marital status, forestry worker, and indigenous people status, to join the SFM program with incentives. Second, the manager can also understand local people's awareness, focusing especially on insights regarding financial support, human capital, and capacity building. Finally, the residents who had higher perceptions on the importance of SFM may be more likely to join the SFM program.

## 5. Discussion and Conclusions

We established an evaluation framework for SFM corresponding to FSC principles based on local people's perspectives using the IPA method, which included workers' rights and employment conditions [44,87], indigenous peoples' rights [12,32], community relations [40,47], environmental values and impact [12,44], and high conservation values [44,52], using eight SFM indicators (Figure 4). Secondly, we analyzed the matrix of the I–P levels of SFM indicators, and overall differences between non-indigenous and indigenous residents' perceptions of the eight SFM indicators, under the IPA evaluation framework [20,23–25,27–29], with solid theoretical constructs. Finally, we identified the factors that affect local people's participation awareness of SFM based on local demographics, perceptions of SFM, and development factors for SFM (i.e., community consensus [19,45,53,57], financial support [19,58], human capital [19,59,60], and capacity building [19,59]). The IPA plot with the improvement index of the SFM provided solid and comprehensive guidelines regarding the policy implications and prioritization of the SFM mechanism from the FSC to capture the perspectives regarding local groups' needs [12,32,40,44,47,87]. The evaluation framework of SFM of this study was conducted with theoretical constructs (Figure 4).

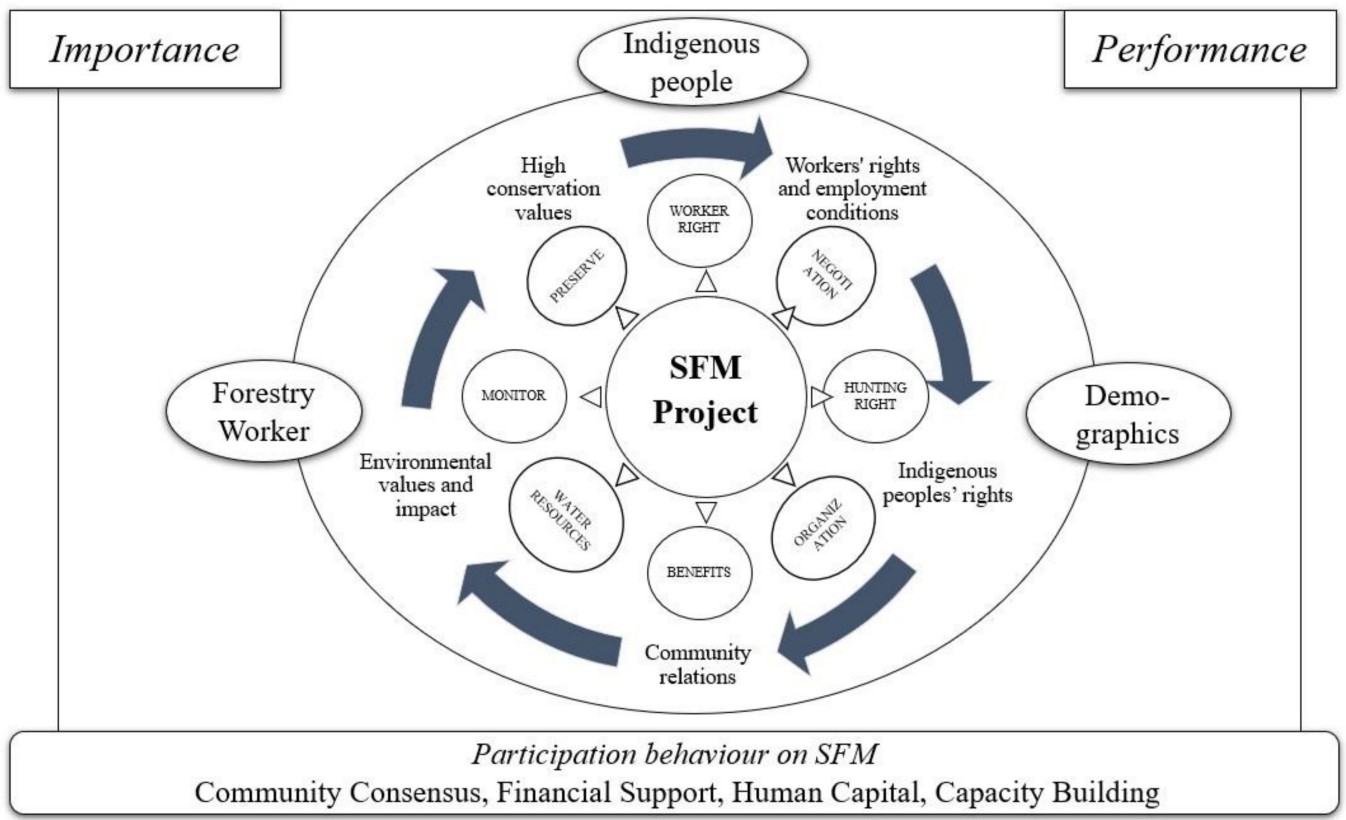

**Figure 4.** The evaluation framework of IPA for SFM in rural area.

The results showed that residents confirmed the high importance of the SFM indicators, while reporting slightly less performance of these indicators. Notable I-P gaps

were detected for all SFM indicators, and local residents generally attached importance to SFM corresponding to FSC principles. All local respondents could explicitly recognize the importance of SFM indicators. Non-indigenous residents, in comparison with the indigenous residents, were more dissatisfied with the importance based on their perceptions of SFM. However, there were similar perceptions between indigenous and non-indigenous residents concerning the SFM performance indicators. Regarding the overall results of the interviewed residents' perceptions of SFM and IPA between indigenous and non-indigenous groups, the following three indicators of SFM that were considered, enhancing the workers' rights for their forest management work, providing the water resources for the local needs, and monitoring the impacts of the SFM and thinning operations on the local environment, were in the A quadrant implying that they should continue to be maintained ("keep up the good work").

The local governance should continue to maintain these advantages and meet the SFM goals of workers' rights and employment conditions [44,87], and environmental values and impact [12,14,44,45]. The development of SFM could improve the workers' benefits and rights on behalf of the local communities [44], and the land rights and employment stability for the SFM [12,87]. Biodiversity and natural resources conservation with a powerful regulatory and monitoring framework is the foundation in SFM for the rural areas [44,50–52]. Holding the SFM and thinning meetings, and negotiating the local people's concern issues proactively for all residents, are located in the B quadrant, which shows that the policymakers should focus on improving indigenous peoples' rights, and pay more attention to this issue [12,14,15]. Thus, community participation of indigenous peoples, as the local stakeholders, is the critical issue for the development of SFM, with regular meetings between local residents and the government sector being essential [12,14,15,32]. However, the indicators of, providing the locals' rights from the hunting and gathering of the forest resources [12,13,32], and providing the social and economic benefits for the local community [12,32,39,42,48], are of lower priority for local management.

Finally, we identified the factors affecting awareness of SFM work based on local people's perceptions through LRM. This study found that community consensus, financial support, human capital, and capacity building, were the main factors for local participation in SFM. The paths towards SFM will not be successful without the local community's participation, especially regarding positive attitudes and relations with SFM [19,45]. Public participation in environmental protection is central to SFM [45,53], along with the long-term adoption of the SFM process [55–57]. The insights regarding public participation imply that it is necessary to consider normative (i.e., affecting decisions and social learning), substantive (i.e., incorporating local knowledge, and containing value-based knowledge), and instrumental (i.e., conflict resolution and reflection) aspects [56], and that key stakeholders are allowed to join the conservation program (i.e., males, forestry workers, and indigenous people) [12–14,65]. Additionally, it is essential to understand the factors influencing the community's participation in SFM. The local people's livelihood dependence on the forests, and increasing their non-forested income, is linked to financial support which is key to SFM [19], to provide a boost to income to enable SFM development [58]. Moreover, providing financial incentives to support non-agricultural employment, reducing the reliance of local communities on forest resources, will encourage the development of SFM [19]. We found that capacity building is the key element for SFM participation, and that adaptability is also core to community capacity building [92]. Community organizations can make local people more resilient by enhancing their skills and knowledge. Thus, locals' ability to adapt to changes in the environment is a key insight for SFM [19,59,92]. Therefore, through training, actively promoted by the government or NGOs, we have established a mechanism for achieving resilience in the ability of community organizations to undertake capacity building [92], achieved through long-term learning and social learning [12,14,65,92]. The role of community participation in generating social learning is often necessary for behavior change and new practice [2,92]. SFM development requires focusing public education on

NRM for human wellbeing and enhanced livelihoods [46,61]. Evidently, SFM cannot be successful without capacity building and human capital.

The main contribution of this research has been to design a measurement indicator for SFM in a rural area of Taiwan. We compared perceptions between indigenous and non-indigenous residents applying IPA among all SFM indicators with a paired sample test, and identified the factors that affect local people's awareness of SFM through LRM. The findings provide both the theoretical constructs and policy implications for the SFM process with the FSC in sustainable rural development. The goals of SFM and rural development can gain insights from this study, which may be designed into an action program for SFM in the future.

## 6. Policy Implication

Most people are unaware that the Hualien Forest Management Office once held a thinning meeting, where not many residents participated. It is recommended that the Forest Service when conducting thinning operations and sustainable forest management, cooperate with relevant experts, scholars, and non-governmental organizations, to hold community seminars and thinning meetings with community residents to discuss community concerns. With a view to gradually seeking consensus and mutual trust, issues affecting thinning operations can then be dealt with by communities and tribes on sustainable forest management issues. The Forestry Bureau and Hualien County Government can cooperate with local experts regarding human resources, funds, education and training, and the legal resources needed by communities and tribes to carry out assistance projects necessary for forest management. Local colleges and universities already have their majors, especially education and training and the understanding of legal sources. Therefore, it is recommended that the Forest Management Office build a cross-departmental integration platform that can identify local needs, government effectiveness, knowledge, and experience transfer.

Valuable indicators for the current forest management system include providing water resources for local needs, and enhancing workers' rights for their forest management work. For example, to provide water resources for forests, due to the sparse population and large living area, in the past there was no corresponding tap water construction, due to factors such as topography and traffic. In recent years, the national policy has also proposed a water supply improvement plan for areas without running water. However, land use is closely related to the demand for water resources, and more accurate monitoring models and controls should be adopted, to maximize the efficiency of forest water use and to reduce the impact on forests. The forest management system attaches great importance to the rights of workers. Since forestry in Taiwan is not for commercial interests, most of the forest management rights of plantation organizations belong to the Forest Service Bureau. This has led to labor problems faced by Taiwan, and the main by-products of international forests. Based on the research results of this study, it is recommended that the Forestry Bureau, and other relevant agencies, consider gradually implementing thinning operations, relaxing the use of primary forest products, and continuing to encourage forest farmers to pursue the under-forest economy, simultaneously carrying out the cultivation and utilization of forest by-products.

The most urgent indicators for improvement of the current forest management system are holding the SFM and thinning meetings, negotiating local people's concern issues proactively, and monitoring the impacts of the SFM and thinning operations on the local environment. During the interviews and survey we also found that local people value the activity of proactively organizing thinning briefings, and presenting the discussion results to the tribes/communities affected by thinning operations. We believe that the Forest Service should increase thinning briefing sessions to encourage two-way communication, and to point out the various impacts of the plan on the environment and ecology. Hence, it is recommended that the Forest Service and other relevant agencies consider the "Principle

3: Indigenous peoples' rights" and "Principle 6: Environmental values and impact" to continue to enable SFM goals.

We found that the forest manager should point out the impact of construction vehicles on the community. It is necessary to increase the number of thinning meetings, and there should be two-way interaction, and the content of the thinning operation should be proposed. The various impacts of the project on the environment and ecology, the rights of indigenous tribes to hunt and use forest water resources, and that there are, for example, cases of garbage disposal in woodlands, etc., should be pointed out. Regarding the consensus of different groups and villages, the above issues are also priority actions that the Forest Management Office can consider. The Forest Management Office should first discuss these consensuses and hold platform meetings with all stakeholders to seek solutions to known and potential action plans. Among different opinions, it is possible to gradually strengthen the connection and communication of information within the tribe, expand the participants of the meeting, and hire local young people with community connections to serve as community patrols. The design and implementation of thinning operation strategy with direct involvement of the local society can support economic interests, while cooperation with tribal elders, county and city governments, and universities, during these projects should strengthen the inheritance of tribal traditional culture and language.

Regarding the promotion of SFM and thinning operations, it is recommended that the forestry authority should plan in the short, medium, and long term for the future: (1) Short-term: use the plan-do-check-action framework for SFM and thinning operations, establish an SFM and thinning operation promotion platform, and carry out various action plans; (2) Medium and long-term: promote various implementation measures based on the principles of FSC. 1. On "labor rights and employment conditions", explore the labor rights and interests aspects of forest management work (such as labor insurance, health insurance, and pay on time), and hire local residents as thinning workers; 2. In "Indigenous peoples' rights", through the SFM and thinning operation promotion platform, regularly hold thinning briefings, continue to exchange various opinions with local stakeholders, and gradually implement various work items; 3. On "community relations", coaching the local communities in the two business areas, setting up community patrol teams, promoting sales of special agricultural products, assisting in the promotion of tribal tourism, and continuing to collect residents' opinions on the promotion of SFM and thinning operations. In order to strengthen community relations and interaction, gradually promote various SFM issues and thinning operations; 4. On "environmental impact assessment", continue to carry out animal and plant monitoring plans based on the monitoring of the past three years, as a reference for the forestry authority to formulate relevant thinning operations management plans; 5. On "maintaining natural forests with high conservation value", it is recommended to hold forest environmental conservation publicity courses, and to communicate with nearby universities, middle, and primary schools. The cooperative environmental education course is expected to gradually implement the concepts and goals of sustainable forest resource management under the principles of FSC.

**Author Contributions:** Conceptualization, K.C., S.S., W.X., V.V.N. and C.-H.L.; Formal analysis, T.-P.T.; Methodology, H.-C.C., S.S., A.E.J.F. and C.-H.L.; Project administration, C.Z.; Resources, T.-P.T. and K.C.; Writing-original draft, H.-C.C., T.-P.T. and C.-H.L.; Writing-review & editing, K.C., A.E.J.F., V.V.N. and C.-H.L. All authors have read and agreed to the published version of the manuscript.

**Funding:** This research was funded by the Ministry of Science and Technology, Taiwan, grant number 108-2410-H-259 -042 and 109-2628-M-259 -001 -MY3. We also thanks for the funder by the Forestry Development Project from Council of Agriculture, grant number 110FD-09.1-HC-02.

**Institutional Review Board Statement:** Not applicable.

**Informed Consent Statement:** Not applicable.

**Data Availability Statement:** The source of illustration in SFM is from the website of MDPI.

**Acknowledgments:** We would like to thank valuable comments from anonymous reviewers, all participants who made this study possible.

**Conflicts of Interest:** The authors declare no conflict of interest.

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
