# Peer review of "Conducting an Evaluation Framework of Importance-Performance Analysis for Sustainable Forest Management in a Rural Area"

_forests, doi:10.3390/f12101357_

Round 1

Reviewer 1 Report

Please see the attached comments.

Author Response

Comments and Suggestions

Response

Major Comments:

The authors conducted a questionnaire to understand the perception of stakeholders towards co-management of forest resources. The findings of the manuscript are well-presented and are important for policymakers. However, research design, results, and conclusion section need to be improved. This manuscript needs to be revised and edited before it becomes publishable in the Forests journal. Finally, I thank the authors and editor for the opportunity to read this work.

Thanks for comments.

We revised the research design, research results, discussions and conclusions based on the suggestions from line 180-198, line 235-242, line 254-267, line 298-312, line 357-365, line 393-403, line 454-472.

Presence of English/grammatical errors and flaws. (for instance, 8 and Eight; repetition of same words such as “moreover, furthermore, thus, and therefore”; too long sentences; and UPPER and lower cases mismatches.

Thanks for the suggestions.

We revised the English/grammatical errors and flaws.

The information need to be organized. Some of the sentences are repeating more than a couple of times.

Thanks for the suggestions.

We revised the repeating sentences.

The methodology needs to be clearly illustrated. How did authors used LRM model? Lack of explanation about nature of explanatory variables.

Thanks for the suggestions.

We added the application of LRM from line 254-267, and also added the explanation on Table 4 in Page 16.

Result is insufficient for third objective.

Thanks for the suggestions.

We revised the “5. Discussions and Conclusions”, and added the discussions for third objective from Line 449-472.

Spell out all the acronyms while appearing at first.

Thanks for the suggestions.

We revised the acronyms while appearing at first.

Introduction

Line 28: Presence of grammatical errors: “…. Not only cannot be……”

Thanks for the reminder.

We revised the grammatical errors in the manuscript.

Line 124-127: “…. thinning operations on the local environment…” what does this sentence means? It is unclear to me.

Thanks for the reminder.

We revised it from Line 131-132.

Line 128-129: Single sentence paragraph is not suitable in the journal.

Thanks for the suggestions.

We revised it from third paragraph in Page 3.

Conceptual framework

The map of the study area is not clear and can be improved.

Thanks for the suggestions.

We revised it in Figure 1, Page 6.

Line 180-182: The reason for selection of study villages is not enough.

Thanks for the suggestions.

We revised it from Line 180-198.

The authors emphasized I-P analysis. However, LRM is overshadowed. I suggest authors to describe LRM with clear explanation of explanatory variables. How variables are created, type of variables (binary or nominal?),

Thanks for the suggestions.

We added a detailed description and model analysis in lines 254-267, and Table 4 in Page 17.

Did authors check the normality?

Thanks for the comments.

We did not check the normality. However, our data and model also fit the normality features as following reason:

1. The formal on-site and random survey for the local residents from July 2019 to March 2020 at our research site throught the Mingli, Dama, Hongye, and Ruixiang villiage surrounding the Gungfu and Ruisui township (Figure 1). Based on the  confidence level of 95%, and estimation bias of 5%, and we assume the indigenous and non-indigenous people had the same preference for the SFM program, we got at a total of 384 samples. Totally, we obtained a total of 400 residents, and a summary of the frequency and percentage for their demographics is shown on Table 2.

2. The results also consistent from two binary choice model (Table 4).The GOF of our empirical model also fit the criterion on the LLR and AIC, which shows that the local people’s participation behavior models of SFM had solid results with model specification under binary choice model, under the normality of Probit model with enough sample size.

3. Above contents were adding in Line 298-302, and Line 393-398.

Results and analysis

Line 243-249: can be improved.

Thanks for the suggestions.

We revised it in lines 305-312.

Table 2: What is the unit of income (monthly/yearly and dollars/which currency)? Is it household income or respondents’ income?

Thanks for the suggestions.

We revised it in Table 2, Page 9.

Line 307: I do not see Table 5 in manuscript.

Thanks for the suggestion.

We corrected it.

In Table 4: The authors have written the prediction accuracy of each model. However, I suggest to write AIC/BIC or ROC for goodness of fits. Also, I think, the objective of the paper is to identify the factors instead of developing the model.

Thanks for the suggestions.

We have added a AIC information in Line 393-398 and Table 4.

Discussions and Conclusions

Insufficient discussion.

Thanks for the suggestions.

We added the discussion from Line 449-472.

Line 354-355: Repeated sentence.

Thanks for the suggestions.

We revised the repeated sentence.

Line 259-371: Need to explain explicitly.

Thanks for the suggestions.

We added the explain from Line 449-472.

Reviewer 2 Report

The manuscript “Conducting an evaluation framework of importance-performance analysis for sustainable forest management in a rural area” presents a conceptual framework of sustainable forest management using indicators closely corresponding with Forest Stewardship Council principles. Work is interesting but in some parts is difficult for the reader.

Remarks

  1. Logistic regression used in the model should be better introduced/described or literature of good description of this method should be cited.
  2. line 180 “… 533 forest classes” – define/ explain cases

Table 1 – In the title of the table give full name instead of the acronym;

Table 2 – give unit for Income – the same in-text

Table 3. Instead of the full name of the indicator repeated three times use the abbreviation introduced in Table 1 – and give information in the footer of the table.

Table 4 – should be rearranged – not clear  - in connection with Model I and model II

The final short conclusion should be formulated – it will improve the whole manuscript.

Author Response

Review Report

Reviewer #2:

Comments and Suggestions

Response

Work is interesting but in some parts is difficult for the reader.

Thanks for the comments.

We revised the research design, research results, discussions and conclusions based on the suggestions from line 180-198, line 235-242, line 254-267, line 298-312, line 357-365, line 393-403, line 454-472.

Logistic regression used in the model should be better introduced/described or literature of good description of this method should be cited.

Thanks for the suggestions.

We added the discussions in line 254-267 and Line 387-403 with citations.

line 180 “… 533 forest classes” – define/ explain cases

Thanks for the suggestions.

We corrected it and add the information from Line 180-190.

Table 1 – In the title of the table give full name instead of the acronym;

Thanks for the suggestions.

We revised it in Page 8, Table 1.

Table 2 – give unit for Income – the same in-text

Thanks for the suggestions.

We revised it in Page 9, Table 2.

Table 3. Instead of the full name of the indicator repeated three times use the abbreviation introduced in Table 1 – and give information in the footer of the table.

Thanks for the suggestions.

We revised it in Page 11, Table 3.

Table 4 – should be rearranged – not clear  - in connection with Model I and model II

Thanks for the suggestions.

We revised it in Page 16, Table 4.

Conclusion should be formulated – it will improve the whole manuscript.

Thanks for the suggestions.

We revised it from Page 17-18.

Round 2

Reviewer 1 Report

Thank you again. The authors have revised/improved the manuscript sufficiently. However, there is still the presence of minor flaws. I suggest authors recheck grammatical errors and tying of methods, results, and discussion.

More specifically, the manuscript also compares the probit and logit model. Please write why manuscripts need both models. Is there any difference in findings? Which one is better off or what? Please write in the methods and results accordingly. 

Suggestions:

  1. Respondents' income: Is it monthly or yearly? Still not clear.
  2. Table 2: Should be the plural form of variable. "Variables" instead of "variable."
  3. Recheck line 230-232.
  4. Typo in Table 4. 
  5. Advised to check font color in Table 4 and References.
  6. Better to write "Marital status" instead of "marriage status" throughout the manuscript. 
  7. Line 349: Grammatical errors in this sentence: "Finally, the residents who had a higher perceptions on importance of SFM also like to join the SFM program." Also, there should be another word instead of "like."